# CAPACITY-LIMITED REINFORCEMENT LEARNING: APPLICATIONS IN DEEP ACTOR-CRITIC METHODS FOR CONTINUOUS CONTROL

## ABSTRACT

Biological and artificial agents must learn to act optimally in spite of a limited capacity for processing, storing, and attending to information. We formalize this type of bounded rationality in terms of an information-theoretic constraint on the complexity of policies that agents seek to learn. We present the Capacity-Limited Reinforcement Learning (CLRL) objective which defines an optimal policy subject to an information capacity constraint. This objective is optimized by drawing from methods used in rate distortion theory and information theory, and applied to the reinforcement learning setting. Using this objective we implement a novel Capacity-Limited Actor-Critic (CLAC) algorithm and situate it within a broader family of RL algorithms such as the Soft Actor Critic (SAC) and discuss their similarities and differences. Our experiments show that compared to alternative approaches, CLAC offers improvements in generalization between training and modified test environments. This is achieved in the CLAC model while displaying high sample efficiency and minimal requirements for hyper-parameter tuning.

## 1 INTRODUCTION

The success of Deep RL has led to significant milestones in AI research (e.g Tesauro (1994), Mnih et al. (2013), Silver et al. (2016)). However, at the same time, some have called into question the extent to which these approaches are able to demonstrate nontrivial generalization of learning, as compared to simply memorizing state-action sequences (Packer et al. (2018), Cobbe et al. (2018)). In this paper, we look to improve sample efficient generalization in RL by making a connection to the idea of capacity limits in the field of Information Theory. As we will demonstrate, this idea has strong theoretical ties to recently popular approaches for Maximum Entropy Reinforcement Learning (MERL) (Haarnoja et al., 2018b) and KL-regularized RL (Tirumala et al., 2019). Furthermore, our new framework for capacity limited RL motivates a new algorithm called Capacity Limited Actor Critic (CLAC) that displays superior out of distribution generalization properties without sacrificing sample efficiency or online learning performance.

RL generally consists of a Markov Decision Process (MDP) defined by a state $s$, an action $a$, a transition function $P(s'|s, a)$ that determines the probability of the next state, and a reward function $r(s, a)$. In this setting the primary objective is typically to learn a policy, which is defined as a probability distribution over actions conditioned on states, $\pi(a|s)$. In fact, we would like to learn a policy that optimizes the expected return over horizon $T$ following the state distribution of the policy $\rho_\pi$: $J(\pi) = \mathbb{E}_{(s_t, a_t) \sim \rho_\pi}[\sum_{t=0}^{T} r(s_t, a_t)]$.

While MDPs define the setting for an idealized learning agent, any physical communication system (biological or artificial) is necessarily limited to transmitting information at a finite rate. In information theory, the information rate of a channel with input $x$ and output $y$ is quantified by the so-called *mutual information*, $\mathcal{I}(X, Y)$. Hence, for any physical information processing system, $\mathcal{I}(X, Y) \leq \mathcal{C}$ for some finite value of $\mathcal{C}$. At the same time, agents with limited information processing capacities should seek to produce behavior that maximizes expected utility. Hence, for a physical agent, optimal behavior is the result of solving a constrained optimization problem (maximizing the utility of behavior, subject to constraints on available information capacity). Recent work in cognitive science has shown that this constrained optimization perspective on information processing can account for

generalization in biological perception (Sims, 2018). In the present work, we apply this perspective to the reinforcement learning framework. In particular, we consider an agent's policy as an information channel, that maps from the current state of the environment, to a probability distribution over actions, and define optimal behavior subject to a constraint on this information capacity of this channel. More formally, an optimally efficient communication channel is one that minimizes expected loss in utility, $\mathbb{E}[L(x, y)]$, to this constraint:

**Goal:** Minimize $\mathbb{E}[L(x, y)]$ w.r.t, $p(x|y)$, subject to $\mathcal{I}(X, Y) \leq \mathcal{C}$

This objective is well-studied within the field of Rate Distortion theory (Tretiak, 1974), a sub-field of information theory. In our application of RD theory onto RL, we consider the policy function $\pi(a|s)$ learned by an RL agent to be a communication channel that maps from the state $s$ onto a probability of performing an action $a$. This allows us to define the optimality condition of an RL agent with a constraint on information representation as

$$\max_{\pi_{0:T}} \mathbb{E}_{(s_t, a_t) \sim \rho_\pi} \left[ \sum_{t=0}^{T} r(s_t, a_t) \right] \text{ s.t } \mathbb{E}_{(s_t, a_t) \sim \rho_\pi} [\mathcal{I}(\pi(a|s))] \leq \mathcal{C}] \; \forall t \tag{1}$$

Where $\mathcal{I}(\pi(a|s))$ is the mutual information of the policy function when taken to be the information channel mapping states onto actions. The introduction of this policy mutual information term in this constraint connects the objective of a RL agent onto the desire of control over the amount of information used to represent agent behaviour. There are multiple ways of calculating this value which will be discussed in the next section. This allows us to define $\mathcal{C}$, the desired maximum channel capacity, and optimize performance in the environment in relation to this capacity. This optimization can be used to define a learning objective that better reflects the reality of information constraints on physical agents, and as we will see, allow for better control over the trade off of immediate performance and generalization.

## 2 CAPACITY LIMITED RL (CLRL)

### 2.1 CAPACITY-LIMITED LEARNING OBJECTIVE

Insight from an information theoretic perspective inspires the goal of maximizing reward obtained subject to some constraints on policy complexity (as measured by its mutual information). In practice, the way we impose a limit on the amount of information that the agent uses to represent its policy is done by applying a penalty to the reward based on this value. This allows us to define a learning objective that regularizes the observed reward:

$$J(\pi) = \sum_{t=0}^{T} \mathbb{E}_{(s_t, a_t) \sim p_\pi} [r(s_t, a_t) - \beta \mathcal{I}(\pi(\cdot|s_t)].$$

The key difference with the standard RL objective the added penalty to the reward observed based on the amount of information that would be required to represent the policy. Policies with higher mutual information values have a greater complexity, in an information-theoretic sense, and this weighted value is used to discourage policies that would require a high information capacity channel. Thus, this learning objective will directly encourage the development of policies that are simple (use low information to represent) but have high utility. Additionally, if there are multiple policies that achieve the same performance, this objective will naturally favor the simplest among them. Higher values of $\beta$ skew the learning objective to prefer policies with less required information.

### 2.2 CALCULATING MUTUAL INFORMATION

The mutual information in the CLRL learning objective can be defined in different ways depending on the application, and the features of the learning environment. Because CLRL describes a broad learning objective that can be applied to different existing RL methods, these different methods may be best suited by different approximations of mutual information. For discrete state and action spaces in tabular learning conditions the space is small enough that it can be defined in terms of the probability mass functions as follows:

$$\mathcal{I}(\pi(a|s)) = \sum_{a \in A} \sum_{s \in S} p_{(s,a)}(s, a) \log \left( \frac{p_{(s,a)}(s, a)}{p_a(a), p_s(s)} \right) \tag{2}$$

However, for more complicated problems, especially environments involving continuous state/action spaces, we will need a different approach. One possibility is to use the definition of mutual information in terms of the probability density functions for continuous distributions. However this method requires the integration over both the state and action spaces, as well as the approximation of both the marginal action $p_a(a)$ and marginal state probabilities $p_s(s)$. An alternative that avoids these complications is to break up the policy mutual information into the components of its constituent entropies. This calculation of mutual information also avoids the approximation of both the marginal state distribution and marginal action distribution, with only one approximation required, being calculated by either of:

$$\mathcal{I}(\pi(a|s)) = \mathcal{H}(\pi_a(a)) - \mathcal{H}(\pi(a|s))$$
$$= \mathcal{H}(\pi_s(s)) - \mathcal{H}(\pi(s|a)) \tag{3}$$

## 2.3 Connections to Maximum Entropy Reinforcement Learning (MERL)

Maximum Entropy RL (Haarnoja et al., 2018a) is a popular RL framework that also makes use of a regularized learning objective to maximize

$$J(\pi) = \sum_{t=0}^{T} \mathbb{E}_{(s_t,a_t)\sim\rho_\pi}[r(s_t,a_t) + \alpha\mathcal{H}(\pi(\cdot|s_t))]$$
$$= \sum_{t=0}^{T} \mathbb{E}_{(s_t,a_t)\sim\rho_\pi}[r(s_t,a_t) - \alpha\mathcal{I}(\pi(\cdot|s_t)) + \alpha\mathcal{H}(\pi(a_t))].$$

This learning objective alters the traditional method by augmenting the reward maximization objective with an additional weighted value based on the entropy of the policy $\alpha\mathcal{H}(\pi(\cdot|s_t))$. This encourages policies that are closer to the uniform distribution and thus more random, since these action distributions will have a higher entropy. The entropy coefficient $\alpha$ controls the weight balancing the reward and entropy of the policy. As this value increases to infinity, the model will learn to always prefer policies that are completely uniform. When this value is 0, the model will only consider the reward returned by the environment and is equivalent to the traditional RL learning objective.

As there is a close theoretical tie between the frameworks, we leverage the successful Maximum Entropy method Soft-Actor-Critic (SAC) as the main point of comparison for our approach. SAC is an off-policy RL method that displays high sample efficiency and resilience to changes in hyperparameters even in difficult continuous control robot simulation environments (Haarnoja et al., 2018b). To make the comparison easier with SAC, we utilize the top equation for the calculation of $\mathcal{I}(\pi(a|s))$ in equation 3. This is because it utilizes the entropy of the policy $\mathcal{H}(\pi(a|s))$, which is calculated in existing MERL methods, instead of $\mathcal{H}(\pi(s|a))$, which represents the difficult to define entropy of the probability of being in a state given that the agent has preformed an action. One benefit to this approach is that any MERL method will naturally need to compute the value $\mathcal{H}(\pi(a|s))$ and can be altered with relative ease to include the additional marginal action term, as is done in this implementation. In the next section on the Deep CLAC algorithm, we apply this alteration to the existing Soft-Actor Critic method that uses the MERL objective to improve exploration. This is done to show that the Capacity-Limited objective can be used to alter existing methods with relative ease, especially ones that already compute the policy entropy $\mathcal{H}(\pi(a|s))$, such as MERL methods.

## 2.4 Connections to KL-Regularized Reinforcement Learning

The KL-Regularized expected reward objective maximizes

$$J(\pi) = \sum_{t=0}^{T} \mathbb{E}_{(s_t,a_t)\sim p_\pi}[r(s_t,a_t) + \alpha\text{KL}(a_t|x_t)]$$

$$\text{KL}(a_t|x_t) = \log\frac{\pi(a_t|s_t)}{\pi_0(a_t|s_t)},$$

referring to the full history up to time t as $x_t = [s_0, a_0, ...s_t, a_t]$. This method is used in the hierarchical learning domain which separates broad high-level task focused decisions and specific

low-level task agnostic actions (Tirumala et al., 2019). In KL-RL, this is done by training a task specific policy $\pi(a|s)$ as well as a default policy $\pi_0(a|s)$ and using the KL divergence between these two policies to regularize the reward observed by the agent. In this way the agent learns to generalize across tasks by increasing the reward observed when reusing learned behaviour across tasks. The default and task specific policies do not converge to the same policy as they have a different level of access to state information, encouraging $\pi_0$ to learn a task independent policy. This difference in state information access can come in the form of additional latent state features available to the task specific policy $\pi$.

The structure of CLAC is not hierarchical and thus we do not compare the performance of CLAC to a KL-RL method, however these two methods do share a mathematical relation that mirrors the connection between KL-RL and SAC. As noted in Tirumala et al. (2019) the maximum entropy objective is a special case of the KL-regularized objective where the default policy is taken to be a uniform distribution. This is intuitively in line with the motivation behind the MERL objective, as it seeks to increase exploration by encouraging random behavior where possible. A similar relation can be made between KL-RL and the CLRL defined here, where the 'default policy' is taken to be the product of the marginal action and state distributions. Because of the motivation of discouraging high information capacity policies, a negative sign is applied to the regularization coefficient giving $\beta = -\alpha$. This derivation relating KL-RL to CLRL is given by first taking the definition of the KL-RL objective:

$$J(\pi) = \sum_{t=0}^{T} \mathbb{E}_{(s_t,a_t) \sim p_\pi}[r(s_t, a_t) + \alpha \mathrm{KL}(a_t|x_t)]$$

$$J(\pi) = \sum_{t=0}^{T} \mathbb{E}_{(s_t,a_t) \sim p_\pi}\left[r(s_t, a_t) - \beta\left[\log \frac{\pi(a_t|s_t)}{\pi_0(a_t|s_t)}\right]\right]$$

We can write the denominator of this logarithm term generally as a function of a state and action such as $\rho(a, s)$. This allows us to define KL-RL to be the case where $\rho_0(a, s) = \pi_0(a_t|s_t)$. In this same manner SAC can be defined as the uniform distribution $\rho(a, s) = 1/||A||$. Finally, CLAC can be defined as $\rho(a, s) = \pi_a(a)\pi_s(s)$ the product of the marginal action and state probabilities.

$$J(\pi) = \sum_{t=0}^{T} \mathbb{E}_{(s_t,a_t) \sim p_\pi}\left[r(s_t, a_t) - \beta\left[\log \frac{\pi(a_t|s_t)}{\pi_a(a_t)\pi_s(s_t)}\right]\right]$$

$$J(\pi) = \sum_{t=0}^{T} \mathbb{E}_{(s_t,a_t) \sim p_\pi}\left[r(s_t, a_t) - \beta \mathcal{I}(\pi(a_t|s_t))\right]$$

As with the relation between SAC and KL-RL, this relation is in line with the motivation of these learning objectives, because this product of marginal state and action functions can be though of as a generic state-independent policy, and we encourage policies that are more similar to this generic policy because they require less information to represent. A clear conceptual example of this is the fact that the information capacity required to represent a policy is minimized by both a completely uniform policy and one that preforms the same action in all states.

## 3 DEEP CAPACITY LIMITED ACTOR CRITIC LEARNING

There are many RL approaches that could be applied in optimizing the capacity-limited learning objective. Here, we implement the CLAC algorithm based on a baseline implementation (Hill et al., 2018) of the Soft Actor-Critic as described in (Haarnoja et al., 2018a). The reason for doing this is to allow for a comparison a CLRL and a MERL objective based method, while keeping other factors such as network structure static. Specifically, this network structure uses policy iteration consisting of a value network with target network updating, 2 q-function networks used to reduce the overestimation bias, and a policy network used to determine the action selection. The derivation of SAC relies on the so-called 'soft value function' $V(s_t) = E_{a_t \sim \pi}[Q(s_t, a_t) + \alpha \mathcal{H}(\pi(a_t|s_t)]$. Here we use the analogous capacity-limited value function $V(s_t) = E_{a_t \sim \pi}[Q(s_t, a_t) - \beta \mathcal{I}(\pi(a_t|s_t)]$.

Because we choose to define the mutual information value in terms of its constituent entropy's, a slight alteration to SAC method is carried forward into defining the gradients of the value function network, q-function networks, and policy network. This allows us to define the gradients used to update these three networks as follows (a full derivation is provided in the appendix),

$$\hat{\nabla}_\chi J_V(\chi) = \nabla_\chi V_\chi(s_t)(V_\chi(s_t) - Q_\kappa(s_t, a_t) - \beta(\log(\pi_\mu(a_t)) - \log(\pi_\phi(s_t, a_t)))),$$

$$\hat{\nabla}_\kappa J_Q(\kappa) = \nabla_\kappa Q_\kappa(s_t, a_t)(Q_\kappa(s_t, a_t) - r(s_t, a_t) - \gamma V_{\hat{\chi}}(s_{t+1})),$$

$$\hat{\nabla}_\phi J_\pi(\phi) = \nabla_\phi \log \pi_\phi(a_t|s_t) + \left(\nabla_{a_t}\beta(\log \pi_\mu(a_t) - \log \pi_\phi(a_t|s_t)) - \nabla_{a_t}Q(a_t, s_t)\right)\nabla_\phi f_\phi(\eta_t; s_t),$$

where $f_\phi(\eta_t; s_t)$ defines the policy under the same reparameterization trick $a_t = f_\phi(\eta_t; s_t)$, used in SAC and KL-RL, where $\eta_t$ is a noise vector sampled from a spherical Gaussian distribution (Haarnoja et al. (2018a)). Together these give the capacity-limited actor-critic algorithm:

---

**Algorithm 1:** Capacity-Limited Actor-Critic

Initialize: parameter vectors $\phi$, $\chi$, $\kappa_1$, $\kappa_2$,
Initialize: Memory $\mathcal{D} = \emptyset$
**for** *each iteration* **do**
    **for** *each environment step* **do**
        $a_t \sim \pi_\phi(a_t|s_t)$
        $s_{t+1} \sim p(s_{t+1}|s_t, a_t)$
        $\mathcal{D} \leftarrow \mathcal{D} \cup \{s_t, a_t, r(s_t, a_t), s_{t+1}\}$
    **for** *each gradient step* **do**
        $\pi_\mu(a_t) \sim \mathcal{D}$ for $a_t \in \mathcal{D}$
        $\chi \leftarrow \chi - \lambda_V \hat{\nabla}_\chi J_V(\chi)$
        $\kappa_i \leftarrow \kappa_i - \lambda_Q \hat{\nabla}_{\kappa_i} J_Q(\kappa_i)$ for $\kappa_i \in 1, 2$
        $\phi \leftarrow \phi - \lambda_\pi \hat{\nabla}_\phi J_\pi(\phi)$
        $\bar{\chi} \leftarrow \tau\chi + (1-\tau)\bar{\chi}$

---

Where $\pi_\mu(a_t) \sim \mathcal{D}$ for $a_t \in \mathcal{D}$ indicates the approximation of the marginal policy function based on an estimation obtained from all actions in the current training mini-batch. This is done by using the batch of actions to create a multi-variate Gaussian distribution of the same order as the action space, and determining the probability of each action given that distribution. Using a different approach to estimating the marginal action probability would be possible, such as training an additional DNN to estimate this marginal and updating it based on observed actions. This would replace the batch approximation step with a gradient update of the marginal action distribution approximation network: $\mu \leftarrow \mu - \lambda_M \hat{\nabla}_\mu J_M(\mu)$.

# 4 LEARNING ENVIRONMENTS

## 4.1 CONTINUOUS N-CHAIN

We explore a continuous action space version of the n-chain environment as described by Strens (2000). This environment consists of $N$ states with agents starting in the $S_1$ state and $S_N$ as the terminal state. Agents act by selecting a continuous value from [0,1], and the probability of them moving to the next state, $p(s_{t+1}|a_t, s_t)$, is proportional to the difference between their action and the hidden value H depending on the state they are in, $H_{s_t}$. These hidden values are sampled from a Beta distribution $Beta(a, b)$. The reward is -1 for all states and 0 for the final state. For all tests shown here the number of states $N = 10$, and beta distribution shape parameters $a = 50$, $b = 50$. In Figure 5 of the appendix we show and graph the state transition function and a diagram for the state and hidden values sampled from the beta distribution.

In Figure 1 we compare the policies learned by CLAC and SAC. For both algorithms, after training we sample 100,000 actions from the policy for a particular state, and plot the resulting probability density function over actions. In this example, the optimal action in this state is $\sim 0.61$. The regularization coefficients for CLAC and SAC (the mutual information term, and the entropy term, respectively) are chosen so that the resulting policies have the same standard deviation of the resulting policy. As shown, the mean of the CLAC action distribution shifts away from the hidden state value towards the across-state mean of the beta distribution. This demonstrates that the CLAC algorithm essentially learns a useful "prior" over its policy representation. Meanwhile, this shift in mean action distribution is not present in the SAC model. This is intuitively related to the difference between these regularization methods as the mean of the distribution relates to the mutual information but not the entropy. We argue that in many natural tasks, policies are are not statistically independent across states, and hence the learning objective of CLAC may better capture useful statistical regularities of the environment, and consequently allow for better learning as well as better generalization

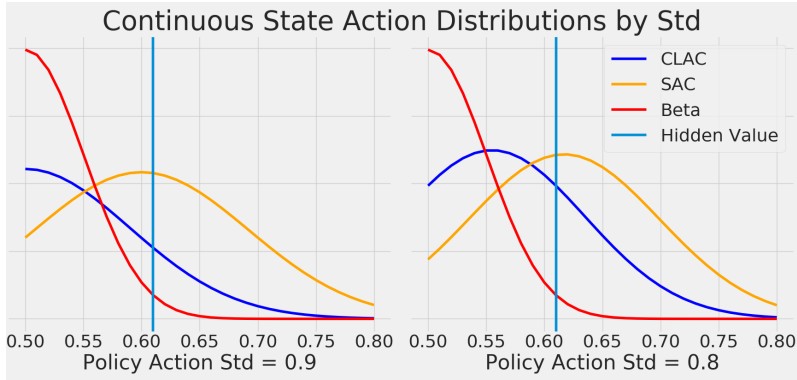

Figure 1: Normal distribution representations of CLAC (blue) and SAC (orange) model policies with standard deviations of 0.8 and 0.9. Light blue displays the value of the hidden state feature defining the state transition probability. Beta distribution used to gereate state features in red.

## 4.2 GENERALIZATION IN CONTINUOUS N-CHAIN

To test the generalizability of the policies learned by CLAC and SAC, these models are first trained on hidden state features generated from one beta distribution, and then placed in an environment with a shifted beta distribution used to generate hidden state features. To ensure that the agents learn the underlying beta distribution and not just a single set of random values, these parameters are re-sampled at the end of each trial. A randomization trial is used where the $b$ parameter of the distribution $Beta(a, b)$ is shifted by $10\%$ from its original. An additional extreme randomization trial is used where the value of $b$ in $Beta(a, b)$ is randomly chosen to be either $+20\%$ or $-20\%$ from the training value. Figure 2 displays these results with the left panel representing the learning environment, the middle representing the randomized environment, and the right panel representing the extreme randomization environment.

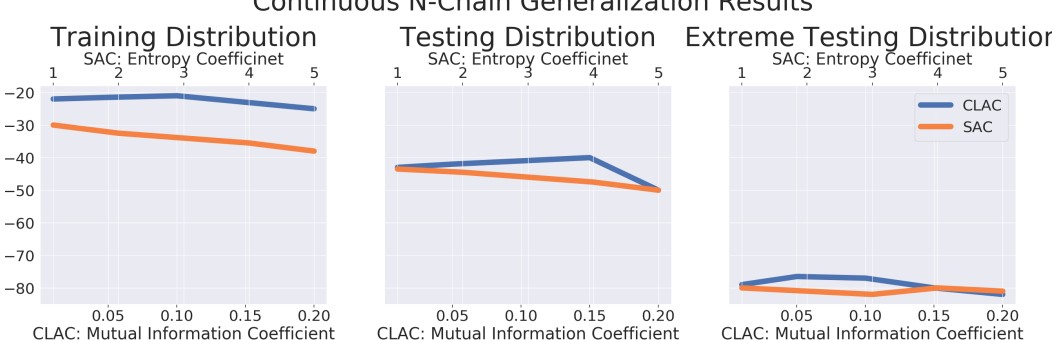

Figure 2: Results from tests in the generalizability of learned policies in SAC and CLAC models, varied by mutual information coefficient and entropy coefficient. Again the number of states N=10, and the number of agents trained is 5. During training, hidden state features are resampled after each trial from the same Beta distribution.

These results show that, for the CLAC model, increasing the value of the mutual information coefficient increases the performance in these perturbed environments. This contrasts the effect of the entropy coefficient in the SAC model, which displays no clear correlation between varying the entropy coefficient and generalized performance. Again this connects to the different motivations behind these models, as the CLAC model enables the shifting of mean actions in a policy as it relates to the policy mutual information, while the mean does not impact the policy entropy and is not captured by the entropy coefficient magnitude.

## 4.3 Continuous Control

In this section we compare the sample efficiency of CLAC against a suite of DRL baselines from the stable baseline implementation (Hill et al. (2018)). The models tested [1] are the Soft Actor-Critic (SAC; Haarnoja et al. (2018a)), Deep Deterministic Policy Gradient (DDPG; Lillicrap et al. (2015)), Proximal Policy Optimization (PPO1; Schulman et al. (2017)), and Advantage Actor Critic (A2C; Mnih et al. (2016)). These specific models were chosen to allow for a fair comparison of the current implementation of the Capacity-Limited Actor-Critic, as opposed to newer methods like PPO2, A3C, and TD3 which leverage improvements such as asynchronous training, parallel workers, or delayed policy update, which are not used in the current CLAC implementation. Although these additional features are not used in the present implementation, they are potential areas of future development in the CLAC method.

As noted, CLAC and SAC are implemented in a similar fashion, and for this reason we compare these models in a series of environments similar to those used to showcase the high sample efficiency and low reliance on hyperparameter optimization of SAC. The learning environments used are the Roboschool environments (Brockman et al. (2016)), an open source version of the same environments [2] used to compare the Soft Actor-Critic method against other models. Figure 6 displays visual representations from these learning environments, ranging from the relatively simple Inverse Pendulum task (top-right) which consists of 1 action dimension and 9 observation dimensions, to the humanoid environment consisting of 17 action dimensions and 44 observation dimensions.

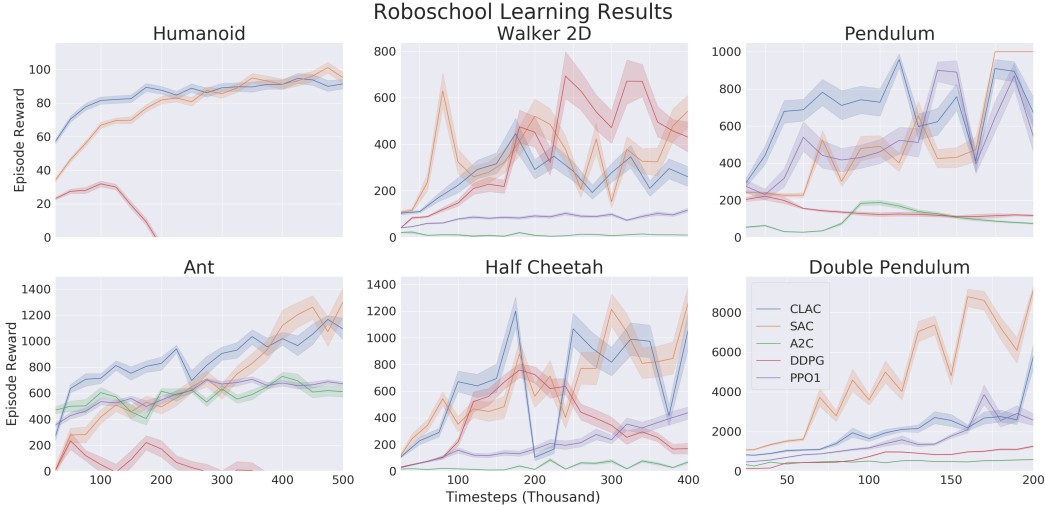

Figure 3: For all tests 5 separate agents are trained. Humanoid and Ant are trained for 0.5M steps, Walker and Cheetah for 0.4 M, and the pendulums for 200K. CLAC mutual information coefficients are chosen to be the optimal values sampled from the range [0,0.3] in 0.025 increments.

The results in Figure 3 indicate that CLAC shares the high sample efficiency of the SAC method, while working with a different learning objective. Additionally, both SAC and CLAC require little hyperparameter optimization. The only parameter that is tuned in the CLAC model is the mutual information coefficient, with the used values for each model included in the Appendix. The SAC model uses an automatic adjustment of the entropy coefficient, that is discussed in the next section. Our CLAC model is implemented using the same policy, state value, and state-action value networks, and these results show that it has retained some key features of SAC while using a different learning objective. This provides support for considering the capacity limited reinforcement learning objective as a general approach that can be applied to different existing RL approaches.

---

[1] All citations for model implementations are taken directly from those referenced by the stable baseline repository https://stable-baselines.readthedocs.io/en/master/

[2] Note that the Roboschool environments have a different reward method from MuJoCu (Todorov et al. (2012)), the environment originally used to test SAC. The Roboschool environment has higher penalties for certain sub-optimal actions, which explains the negative performance in for some models in some environments.

## 4.4 Generalization in Continuous Control

To test generalization in continuous control environments, we utilize the standard OpenAI Gym Pendulum environment, a similar task as the Roboschool Pendulum environment, which will allow us to vary environment features like the mass and length of the pendulum. We adapt the specific method of generating randomized environment features from Packer et al. (2018), by varying both the mass and length of the pole in the Pendulum task from a given range after each episode of testing. In the training parameter case, the mass and length is kept the same as was used during training, equal to 1. In the testing parameter case the pole mass and length are both sampled randomly from the disjoint set, $[0.5, 0.7] \cup [1.3, 1.5]$. In the extreme testing parameter case, the features are sampled from a disjoint set further from the training values: $[0.3, 0.5] \cup [1.5, 1.6]$.

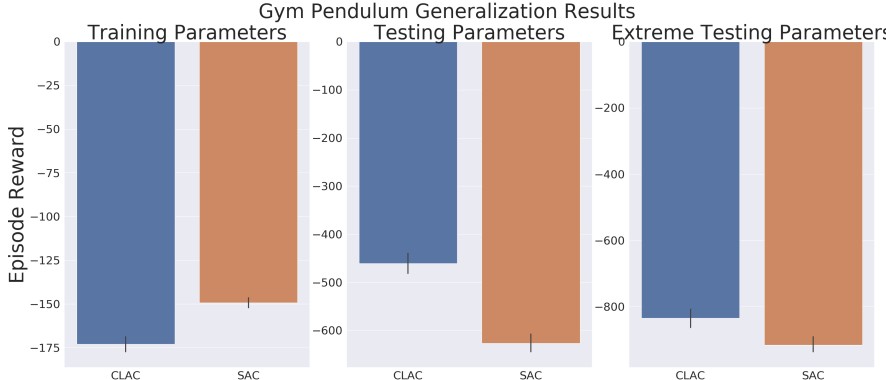

Figure 4: Results from tests in the generalizability of learned policies in SAC and CLAC models with 5 agents each. Static parameters represent the same environment features used to test agents. Testing parameters represent sampling from a disjoint set around the training parameters. Extreme testing parameters represent a disjoint set further from the training than the random parameters. Error bars represent 95% confidence interval.

These results show the improved generalization of the policies learned by the CLAC model compared to Soft-Actor Critic in a continuous control task. Because of the complexity of the task compared to the more simple continuous n-chain environment, this improvement in generalization comes at a cost to training performance. SAC and CLAC models were trained for the same number of time steps before testing their generalization capability. This experiment did not compare the performance to other models like DDPG, PPO, and A2C because their lower sample efficiency meant that their policies at the same time in training were too poor to preform well in even the static parameter case.

## 5 Automatic Mutual Information Coefficient Adjustment

In defining the algorithm for encouraging policies that required less information to represent, we took the practical approach of applying a penalty to the reward observed by an agent based on the complexity of the agents policy. However, there is a way that we can regain our original objective of maximizing reward subject to a given policy channel capacity. This is done by defining a target mutual information for our policy, and automatically updating our coefficient to approach that desired target while training. This is done much in the same way as the automatic adjustment of the entropy coefficient $\alpha$ used in SAC that is described in Haarnoja et al. (2018b). A full derivation of this method is provided in the Appendix. This derivation provides us the following gradient used to update the value of $\beta$:

$$J(\beta) = \mathbb{E}_{a_t \sim \pi_t}[\beta(\log \pi_t(a_t|s_t) - \log \pi_t(a_t|s_t)) + \beta \bar{\mathcal{C}}]$$

where $\mathcal{C}$ is the desired maximum policy channel capacity. Thus we can define the automatic mutual information coefficient adjustment version of Algorithm 1, as including the update:

$$\beta \leftarrow \beta - \gamma \hat{\nabla}_\beta J(\beta)$$

As we have seen, the mutual information coefficient controls the extent to which the CLRL algorithm prioritizes some degree of policy generalizability over training reward maximization. However, as with the entropy coefficient in SAC, this mutual information coefficient is sensitive to the scale of the reward of the learning environment, because the penalty $-\beta \mathcal{I}(\pi(a|s))$ is applied to the reward observed in the environment. Automatically adjusting the entropy coefficient $\alpha$ is partly motivated by invariance to reward scale, and the same benefit is gained by automatically adjusting the mutual information coefficient in CLAC.

However with CLAC, this automatic adjustment has the additional connection to the original motivation for applying a capacity-limit, to allow us to define a channel capacity and maximize reward relative to it. Another benefit of this approach is mitigating some of the potential issues caused by negative transfer, a difficult open question in the area of generalization where attempting to transfer learning can negatively impact performance (Taylor & Stone (2009)). These negative effects could be mitigated by having the model update the $\beta$ coefficient throughout learning and determine which value of the coefficient best reflects the degree to which knowledge of a policy in one state can be applied to other states. Only tight constraints on information capacity should be disastrously impacted by negative transfer, as weaker constraints do not force agents to reuse policies in states where they are not associated with rewards.

## 6 DISCUSSION

In this work we present a formalization of the Capacity-Limited Reinforcement Learning (CLRL) objective and relate it to existing methods in Deep Reinforcement Learning. We argue that CLRL defines a broad approach that can be applied onto existing RL methods to provide better of control over some aspects of generalization. To support this position we present the the Capacity-Limited Actor-Critic (CLAC), an application of CLRL onto a deep off-policy actor-critic model. We use a continuous N-chain environment to clarify the impact on performance and generalization afforded by altering parameters of the CLAC model, specifically the mutual information coefficient.

Empirical results from robot simulation tasks show that the CLAC model can achieve similar sample efficiency and performance as state-of-the-art methods in complex tasks while using the CLRL objective. This is a key result as this objective differs from that of the similar Maximum Entropy learning objective. MERL based methods use an entropy coefficient to encourage exploration through more random behaviour where possible. Meanwhile, CLRL based methods use the mutual information coefficient to balance one aspect of generalization, and is motivated by the natural capacity for storing and processing information that exists in biological agents. This difference present in CLAC enabled it to outperform existing methods in environments with perturbed features, displaying the improvement in generalization afforded by a capacity-limited learning objective.

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

# 7 APPENDIX

## 7.1 GRADIENTS DERIVATION

Because of the mathematical connection to the maximum entropy learning objective present in the capacity-limited learning objective, we are able to derive the gradients for training the three DNNs employed by our agent much in the same manner as is used by the original SAC paper (Haarnoja et al. (2018a)). This alteration is particularly straightforward due to our use of the entropy based definition for mutual information. To achieve this derivation, we replace the use of the soft q-function and v-function in SAC with the capacity-limited q-function and v-function, and carry forward this alteration into the remainder of the derivation.

Taking the capacity-limited value function

$$V_\chi(s_t) = \mathbb{E}_{a_t \sim \pi}\left[Q(s_t, a_t) - \beta(\log \pi_\mu(a_t) - \log \pi(a_t|s_t))\right]$$

With state independent marginal action function $\pi_\mu(a_t)$. We train the agent to minimize the squared residual error between our capacity-limited value function and $Q_\kappa(s_t, a_t)$:

$$J_V(\chi) = \mathbb{E}_{s_t \sim D}\left[\frac{1}{2}\left(V_\chi(s_t) - \mathbb{E}_{a_t \sim \pi_\phi}\left[Q_\kappa(s_t, a_t) + \beta\big(\log \pi_\mu(a_t) - \log \pi_\phi(a_t|s_t)\big)\right]\right)^2\right]$$

which can be optimized via stochastic gradients with:

$$\hat{\nabla}_\chi J_V(\chi) = \nabla_\chi V_\chi(s_t)(V_\chi(s_t) - Q_\kappa(s_t, a_t) - \beta(\log(\pi_\mu(a_t)) - \log(\pi_\phi(s_t, a_t))))$$

Taking the capacity-limited state-value function:

$$\hat{Q}(s_t, a_t) = r(s_t, a_t) + \gamma \mathbb{E}_{s_{t+1} \sim \rho}[V_\chi(s_{t+1})]$$

where $V_\chi(s_t)$ is the capacity-limited value function, trained to minimize the Bellman residual:

$$J_Q(\kappa) = \mathbb{E}_{(s_t, a_t) \sim \mathcal{D}}\left[\frac{1}{2}\big(Q_\kappa(s_t, a_t) - \hat{Q}_\kappa(s_t, a_t)\big)^2\right]$$

again optimized via stochastic gradient descent.

$$\hat{\nabla}_\kappa J_Q(\kappa) = \nabla_\kappa Q_\kappa(s_t, a_t)(Q_\kappa(s_t, a_t) - r(s_t, a_t) - \gamma V_{\hat{\chi}}(s_{t+1}))$$

We begin with the same policy updating method as in SAC, and then replace the soft functions with the capacity-limited functions. This results in a guarantee on improved performance in terms of the capacity-limited value objective. We constrain $\pi \in \Pi$ where $\Pi$ represents the spherical Gaussian family. At each step we update the policy according to

$$\pi_{\text{new}} = \arg\min_{\pi' \in \Pi} D_{KL}\left(\pi'(\cdot|s_t) \big|\big| \frac{\exp(Q^{\pi_{\text{old}}}(s_t, \cdot))}{Z^{\pi_{\text{old}}}(s_t)}\right)$$

After employing the policy reparameterization trick used in SAC and KL-RL we can rewrite this objective as:

$$J_\pi(\phi) = \mathbb{E}_{s_t \sim D}\left[\mathbb{E}_{a_t \sim \pi_\phi}[\beta\big(log(\pi_\mu(a_t)) - log(\pi_\phi(a_t|s_t))\big) - Q_\kappa(a_t, s_t)]\right]$$

which gives the gradient as:

$$\hat{\nabla}_\phi J_\pi(\phi) = \nabla_\phi \log \pi_\phi(a_t|s_t) + \big(\nabla_{a_t}\beta(\log \pi_\mu(a_t) - \log \pi_\phi(a_t|s_t)) - \nabla_{a_t}Q(a_t, s_t)\big)\nabla_\phi f_\phi(\eta_t; s_t)$$

## 7.2 Continuous N-Chain Environment

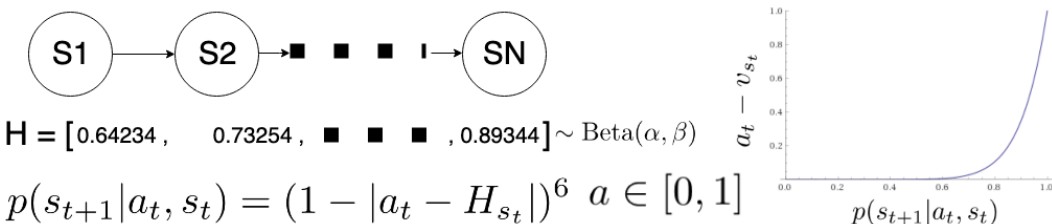

$$H = \begin{bmatrix} 0.64234, & 0.73254, & \blacksquare & \blacksquare & \blacksquare & , 0.89344 \end{bmatrix} \sim \text{Beta}(\alpha, \beta)$$

$$p(s_{t+1}|a_t, s_t) = (1 - |a_t - H_{s_t}|)^6 \ a \in [0, 1]$$

Figure 5: Top: Diagram of the Continuous N-Chain Learning environment. Middle: Example of a set of hidden state values. Right: Graph of the probability of moving to the next state given the absolute distance from the hidden state value and action preformed. Bottom: Function describing this state transition probability.

## 8 Roboschool Environments

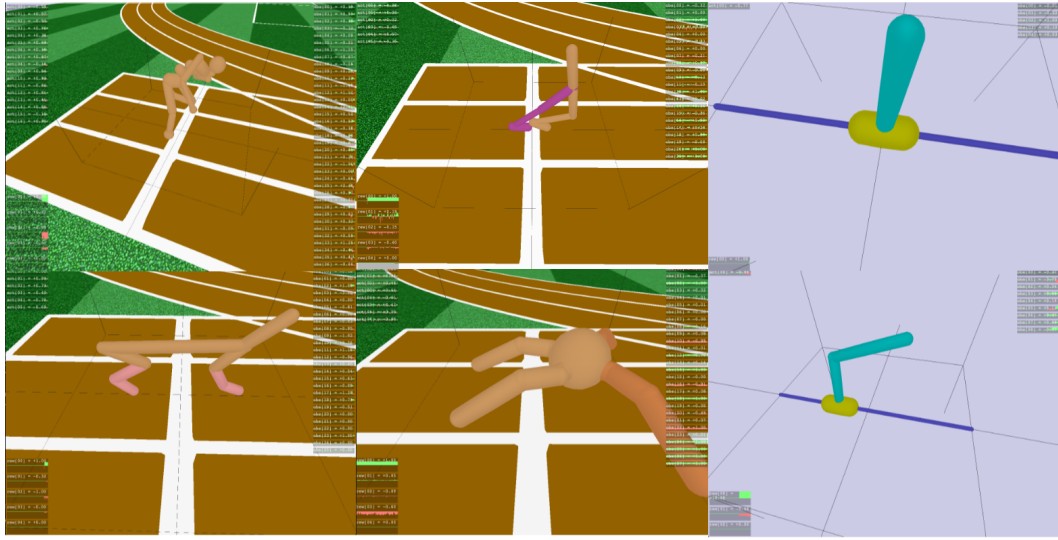

Figure 6: Images of the six roboschool environments used to compare performance CLAC against existing Deep RL methods. Top-left: humanoid. Top-middle Walker 2D. Top-right Inverted Pendulum. Bottom-left: Half-Cheetah. Bottom-middle: Ant Walker. Bottom-right: Inverted Double Pendulum.

## 8.1 Roboschool Mutual Information Coefficients

Table 1: Model Hyper-parameters

| Environment | Mutual Information Coefficient |
|---|---|
| Humanoid | 0.075 |
| Walker 2D | 0.025 |
| Half Cheetah | 0.075 |
| Ant | 0.15 |
| Inverted Pendulum | 0.1 |
| Double Inverted Pendulum | 0.3 |

## 8.2 AUTOMATIC COEFFICIENT UPDATE DERIVATION

To derive this approach, Haarnoja et al. (2018b) define the MERL constrained optimization problem as:

$$\max_{\pi_{0:T}} \mathbb{E}_{\rho_\pi} \left[ \sum_{t=0}^{T} t(s_t, a_t) \right] \text{ s.t } \mathbb{E}_{(s_t,a_t)\sim\rho_\pi}[-\log(\pi(a,s))) \leq \mathcal{H}] \ \forall t \tag{4}$$

Where $\mathcal{H}$ is the desired minimum target entropy. Using a dynamic programming approach and the recursive definition the MERL soft-Q function, this constrained optimization problem produces the optimal dual variable $\alpha_t^*$

$$\alpha_t^* = \arg\min_{\alpha_t} \mathbb{E}_{\alpha_t}[-\alpha_t \log(\pi_t^*(a_t|s_t; \alpha_t)) - \alpha_t \bar{\mathcal{H}}] \tag{5}$$

which is calculated in the practical application through gradient descent according to the objective:

$$J(\alpha) = \mathbb{E}_{a\sim\pi_t}[-\alpha \log \pi_t(a_t|s_t) - \alpha\bar{\mathcal{H}}] \tag{6}$$

We begin the derivation of the similar automatic adjustment for the mutual information coefficient by using the entropy definition of the mutual information $\mathcal{I}(\pi(a|s)) = \mathcal{H}(\pi(a)) - \mathcal{H}(\pi(s|a))$ applied to Eq. 1 as follows:

$$\max_{\pi_{0:T}} \mathbb{E}_{\rho_\pi} \left[ \sum_{t=0}^{T} t(s_t, a_t) \right] \text{ s.t } \mathbb{E}_{(s_t,a_t)\sim\rho_\pi}[-(log(\pi(a) - log(\pi(a,s))) \leq C] \ \forall t \tag{7}$$

Because our capacity-limited value function is recursively defined in the same manner as in SAC, and we are using the entropy based definition of the policy mutual information, we can define the optimal dual variable and use a gradient descent method to update this coefficient during learning, giving:

$$J(\beta) = \mathbb{E}_{a_t\sim\pi_t}[\beta(\log \pi_t(a_t|s_t) - \log \pi_t(a_t|s_t)) + \beta\bar{\mathcal{C}}] \tag{8}$$

where $\mathcal{C}$ is the desired maximum policy channel capacity. Thus we can define the automatic mutual information coefficient adjustment version of Algorithm 1, as including the update:

$$\beta \leftarrow \beta - \gamma\hat{\nabla}_\beta J(\beta) \tag{9}$$

at each gradient step. In the results section we compare the performance of CLAC against SAC using automatic coefficient adjustment as well as a constant coefficient using the static coefficient with the best performance.

## 8.3 MODEL HYPER-PARAMETERS

| Parameter | Value |
|-----------|-------|
| *All Models* | |
| Optimizer | Adam |
| Learning Rate | 3e-4 |
| Discount($\gamma$) | 0.99 |
| Replay Buffer Size | 256 |
| Number of Hidden Layers | 2 |
| Number of Nodes per Layer | 256 |
| Minibatch Size | 256 |
| | |
| *CLAC and SAC* | |
| Nonlinearity | ReLu |
| *Other Models* | |
| Nonlinearity | Tanh |