# OpenReview forum: "CAPACITY-LIMITED REINFORCEMENT LEARNING: APPLICATIONS IN DEEP ACTOR-CRITIC METHODS FOR CONTINUOUS CONTROL"
_ICLR.cc/2020/Conference — Reject_

### Official Review · AnonReviewer2 · 2019-10-18
**Official Blind Review #2**

**Rating:** 3

**Review:**

The paper presents a reinforcement learning method that regularizes the objective using the mutual information term.
The idea is simple and the paper is easy to follow.

However, the novelty is limited since the difference between the proposed method and Soft Actor Critic (SAC) is just adding the entropy term of \pi(a) to the objective function if I understand the method correctly. In addition, the intuition of adding the entropy term of \pi(a) to the objective is not clearly described.

The proposed method is evaluated on continuous control tasks.The results shown in the paper is mixed, and I cannot conclude that the proposed method outperforms SAC. Thus, the benefit of the proposed method is not clearly supported by the experimental results.

For the current form of the paper, I give "weak reject" due to the weak support of the experimental results and the unclear motivation of the method.

One of my concerns is that the way of estimating the \pi(a) which is the marginal distribution of the action.
From the current manuscript, I did not fully understand how it is estimated in the proposed method.
I think that the accuracy of the estimation of \pi(a) is crucial in the proposed method since it is the difference from SAC.

A comment on the paper structure is that the connection to the "capacity-limited" objective should be described more explicitly in Section 2.1.
Although the "capacity-limited" reminds me of the objective something like \mathcal{L} + | C - I(X;Y)  | as in [Dupont, 2018],
the objective in the proposed method shown in page 2 is \mathcal{L} + \beta I(X;Y).
I did not understand why the proposed method is "capacity-limited" until Section 5.
I think authors should explicitly mention in Section 2.1 that \beta is adjusted so as to limit the information capacity.

To improve the manuscript, I request authors the following things:

- The proposed method can be interpreted as adding the penalizing the entropy of \pi(a) to the entropy-regularized RL.
I do not fully understand the intuition of penalizing the entropy of \pi(a) in the context of RL. Please explain it.

- I think some tasks should be performed with longer training.
For example, agent should be trained for 1-2 millions steps on Humanoid and Ant tasks.
In addition, evaluation with 10 trials are preferable.

- Please cite papers that estimate the marginal distribution \pi(a) in the same manner.
If there is no previous work, please explain the details of estimating \pi(a) and how \pi(a) is approximated from samples.


**Experience Assessment:**

I have published one or two papers in this area.

**Review Assessment: Checking Correctness Of Derivations And Theory:**

I assessed the sensibility of the derivations and theory.

**Review Assessment: Checking Correctness Of Experiments:**

I assessed the sensibility of the experiments.

**Review Assessment: Thoroughness In Paper Reading:**

I read the paper at least twice and used my best judgement in assessing the paper.

---

### Official Review · AnonReviewer1 · 2019-10-23
**Official Blind Review #1**

**Rating:** 1

**Review:**


# Summary
The paper replaces the entropy term in the objective of SAC with a KL(\pi(a|s) || p(a)) where p(a) is the marginal distribution p(a) = \int \pi(a|s) \mu(s) ds. It derives the gradient of this modified objective and then proceeds to evaluate the resulting "SAC with mutual information" to demonstrate (i) better sample efficiency and (ii) generalization to environment change.


# Decision
The paper proposes an interesting approach but a more thorough evaluation is needed before it can be recommended for publication.


# Comments
The theoretical contribution is quite small since the connection to RD theory is well established by now, e.g., see papers by Daniel Alexander Braun from Ulm and his students. Therefore, the main contribution is incorporation of the mutual information into SAC. Then the question is what properties/advantages/disadvantages such approach brings. And on this front, the paper is quite weak.

The chain environment is quite toy-ish and may only serve to indicate that the implementation is correct. The continuous control environments are more interesting, but the learning curves look very unreliable. Just a quick Google search for SAC results on Roboschool reveals much smoother and higher learning curves (e.g., https://medium.com/@kengz/soft-actor-critic-for-continuous-and-discrete-actions-eeff6f651954). It would be paramount to make sure that the baselines are fairly represented in the evaluations.

The generalization in continuous control is only evaluated on the pendulum by varying length and mass. It is insufficient to make a decisive judgement. Moreover, length and mass are coupled, such that different combinations of length and mass may yield similar dynamics.

=> If better sample efficiency and better generalization are claimed, then a more thorough evaluation is required.

In general, writing can be improved, figure made nicer and smaller, introduction and connections section made shorter, and the main derivation moved from the Appendix into the main body and explained better.


**Experience Assessment:**

I have published one or two papers in this area.

**Review Assessment: Checking Correctness Of Derivations And Theory:**

I did not assess the derivations or theory.

**Review Assessment: Checking Correctness Of Experiments:**

I assessed the sensibility of the experiments.

**Review Assessment: Thoroughness In Paper Reading:**

I read the paper at least twice and used my best judgement in assessing the paper.

---

### Official Review · AnonReviewer3 · 2019-10-24
**Official Blind Review #3**

**Rating:** 1

**Review:**

The authors propose capacity-limited reinforcement learning and apply an actor-critic method (CLAC) in some continuous control domains. The authors claim that CLAC gives improvements in generalization from training to modified test environments, and that it shows high sample efficiency and requires minimal hyper-parameter tuning.

The introduction started off making me think about this area in a new way, but as the paper continued I started to find some issues. To begin with, I think the motivation in the introduction could be improved. Why would I choose to limit capacity? This is not sufficiently motivated. I suspect that the author(s) want to argue that it *should* give better generalization, but this argument is not made very clearly in the introduction. Perhaps this is because it would be difficult to make this argument formally, and so it is merely suggested at?

Are there connections between this and things like variational intrinsic control (VIC, Gregor et al. 2016) and diversity is all you need (DIAYN, Eysenbach et al., 2019)? These works aim to maximize the mutual information between latent variable policies and states/trajectories, whereas this work is really doing the opposite. I would be interested in understanding the author’s take on how the two are related conceptually.

Moving to the connections with past work, this paper seriously abuses notation in a way that actually hinders comprehension. Some of the parts that really bothered me, and should be fixed to be correct:

Mutual information is a function of two random variables, whereas it is repeatedly expressed as a function of the policy. Being explicit about the random variables / distribution here is pretty important.

In Equation 2 (and subsequent paragraph) the marginal distributions p_a(a) and p_s(s) are not well defined, marginalizing over what, what are these distributions? I might guess that p_s(s) is the steady state distribution under a policy pi, and that p_a(a) is marginalizing over the same distribution, essentially capturing the prior probability of each action under the policy. But these sort of things need to be said explicitly.

In KL-RL section there is a sentence with “This allows us to define KL-RL to be the case where p_0(a, s) = \pi_0(a_t | s_t).” What does this actually mean? One of these is a joint probability for state and action, and one is an action probability conditional on a state.

What does \pi_\mu(a_t) \sim \mathcal{D} mean?

In the block just before Algorithm 1, many of these symbols are never defined. This needs a significant amount of care (by the authors) and right now relies on the reader to simply make a best guess at what the authors probably intend.

Overall in the first three sections the message I would like the authors to understand is that, in striving for a concise explanation they have significantly overshot. These sections require some significant work to be considered publishable.

The experiment in section 4.1 is intended to give a clean intuitive understanding of the method, but falls a bit short here. It is clean, but I needed more explanation to really drive the intuition home. I see that CLAC finds a solution more sensitive to the beta distribution, but help me understand why this is the right solution in this particular case.

I really disagree with the conclusions around the experiments in section 4.2. I do not think these results show that for the CLAC model increasing the mutual information coefficient increases performance on the perturbed environments. First, the obvious, how many seeds and where are the standard deviations? Second, the trend is extremely small and the gap between CLAC and SAC is just as minor. Finally, CLAC has better performance on the training distribution which means that it actually lost *more* performance than SAC when transferring to the testing and extreme testing distributions.

The results for section 4.3 are just not significant enough to draw any real conclusions. The massive temporal variability makes me very suspicious of those super tight error bands, but even without that question, the gap is just not very large.

Finally, in section 4.4 we see the first somewhat convincing experimental results. These look reasonable, but even here I have a fairly pointed question: compared with the results in Packer et al (2018) the amount of regression from training to testing is extremely large (whereas they found vanilla algorithms transfer surprisingly well). Can you explain why there is such a big discrepancy between those results and these? But again, this section’s results are in my opinion the most convincing that something interesting is happening here.

Lastly, in section 8.1 the range of hyper-parameters for the mutual information coefficient is very broad, which really makes it hard to buy the claim of requiring minimal hyper-parameter tuning.

All in all there is something truly interesting in this work, but in the present state I am unable to recommend acceptance, and the amount of work required along with questions raised lead me to be fairly confident in this assessment.


**Experience Assessment:**

I have published one or two papers in this area.

**Review Assessment: Checking Correctness Of Derivations And Theory:**

I assessed the sensibility of the derivations and theory.

**Review Assessment: Checking Correctness Of Experiments:**

I carefully checked the experiments.

**Review Assessment: Thoroughness In Paper Reading:**

I read the paper at least twice and used my best judgement in assessing the paper.

---

### Decision · Program_Chairs · 2019-12-19

**Decision:**

Reject

**Comment:**

This paper presents Capacity-Limited Reinforcement Learning (CLRL) which builds on methods in soft RL to enable learning in agents with limited capacity.

The reviewers raised issues that were largely around three areas: there is a lack of clear motivation for the work, and many of the insights given lack intuition; many connections to related literature are missing; and the experimental results remain unconvincing.

Although the ideas presented in the paper are interesting, more work is required for this to be accepted. Therefore at this point, this is unfortunately a rejection.